# Physicochemical, Structural, and Functional Properties of Snake Melon (*Cucumis melo* subsp. *melo* Var. *flexuosus*) Microencapsulated with Pea Protein and Pea Fibre by Freeze-Drying

**DOI:** 10.3390/foods12142679

**Published:** 2023-07-11

**Authors:** Marta Igual, Alejandro Flores-León, Belén Picó, Javier Martínez-Monzó, Purificación García-Segovia

**Affiliations:** 1i-Food Group, Instituto Universitario de Ingeniería de Alimentos-FoodUPV, Universitat Politècnica de València, Camino de Vera s/n, 46022 Valencia, Spain; xmartine@tal.upv.es (J.M.-M.); pugarse@tal.upv.es (P.G.-S.); 2Instituto de Conservación y Mejora de la Agrodiversidad Valenciana, Universitat Politècnica de València, 46022 Valencia, Spain; alfloleo@doctor.upv.es (A.F.-L.); mpicosi@btc.upv.es (B.P.)

**Keywords:** snake melon, powders, bioactive compounds, antioxidant capacity, encapsulation efficiencies

## Abstract

The purpose of this study was to obtain a functional, stable powder product from *Cucumis melo* subsp. *melo* Var. *flexuosus* (L.) to promote its consumption and reduce waste and production losses. The melons were ground and freeze-dried with or without biopolymers (pea protein (PPSM) or pea fibre (PFSM)). The physicochemical, structural, and functional properties of the powder were studied. The water content, water activity, bulk density, porosity, Hausner ratio, Carr index, hygroscopicity, water solubility, water absorption index, particle size, colour, and microstructure of the powder were determined. In addition, vitamin C, folates, chlorophyll a, total phenols and carotenoids, antioxidant capacity, and powder encapsulation efficiency were analysed. Snake melon (SM) powders contained vitamin C, folates, carotenoids, chlorophyll a, and phenols, which contributed to their antioxidant capacity. The incorporation of PP or PF in the formulation before lyophilisation generated stable encapsulates that protected the bioactive compounds. PPSM and PFSM were less hygroscopic and more free-flowing and had lower water content and water activity compared to the SM. PFSM showed higher encapsulation efficiency and smaller particles with a smooth surface and oval shape.

## 1. Introduction

Melon (*Cucumis melo* L.) is a morphologically diverse crop with a wide range of polymorphic fruit types [1] belonging to the Cucurbitaceae family, which also includes several other important vegetables, such as watermelon, gourds, pumpkin, and cucumber [2]. Traditionally, in the Mediterranean region, snake melon (*Cucumis melo* L. subsp. *melo* Var. *flexuosus* (L.)) (SM), a singular type of nonsweet melon, has been cultivated since Roman times, as described by Roman authors Columella and Pliny the Elder [3]. This crop remains important in different areas worldwide, not only in the Mediterranean Basin but also in North Africa, the Middle East, and Asia Minor, where it is known under different local denominations: Fakous, Kakri, and Armenian cucumbers, among others [4]. This crop is still produced and appreciated in Spain, especially in “Levante” (eastern coastal Spain), where it is locally known as “alficoz” or “alficoç”. Due to its early flowering and short growing cycle, it is being tested as a crop suitable for organic agriculture as it can overcome soil-borne problems that limit the cultivation of sweet melons [5].

This vegetable is often eaten at the immature fruit stage and is consumed both fresh and cooked, like cucumbers, as the nonsweet fruits become inedible when they are mature [6,7]. One problem with SMs is their short shelf life [8]. To avoid this, studies have investigated if this waste can be reduced and how to promote its consumption [9]. Other studies have proposed using snake melon seeds for oil as they have good colour and appearance and high nutritional value [10,11]. As for the fruits, SMs have been analysed in previous studies [12,13], which showed low dietary fibre and vitamin C, carotenoids, and phenols contents.

SM powders could solve problems related to the short shelf life of fruits and their seasonality and provide microbiologically stable products because of their low water activity [14,15]. Powder products facilitate shipping operations and increase profitability because of their lower volume and weight and easier handling. Freeze-drying has emerged as a drying method that generates high-quality products with very low moisture content, good sensory and nutritional properties, and good capacity for rehydration. This method provides the highest chemical profile retention and antioxidant activity in foods, which are attributed to its less intense heating [16]. Furthermore, SM powder could be used as an ingredient for preparing other foods, to season salads, or to flavour ice cream or snacks [14,15,17]. Using food powders could promote SM cultivation and consumption and considerably reduce waste. The fruit of the SM, with its peculiar snake shape, is difficult to transport and can be easily damaged. This proposal would avoid these losses. To obtain freeze-dried vegetables, it is important to use high-molecular-weight additives with the product before drying, such as carrier and anticaking agents, to stabilise hygroscopic powders [18,19]. Pea protein is currently used as an encapsulating agent in both spray-drying [20] and freeze-drying [21] with high encapsulation efficiency. There are also incipient studies incorporating pea fibre with satisfactory results [22]. Using biopolymers improves and maintains the characteristics of powder products but also allows the microencapsulation of bioactive compounds in the matrix [23,24,25]. Microencapsulation provides a physical barrier around the target compounds, reducing the contact and reactivity of the encapsulated material with the environment [26]. Consequently, microencapsulation has been found to be an excellent tool for stabilising bioactive compounds [27,28] and including compounds in food matrices as food ingredients [26].

The main goal of this study was to obtain a structural, functional, stable powder product from *C. melo* L. subsp. *melo* Var. *flexuosus* (L.) Naudin to promote its consumption and reduce waste and production losses.

## 2. Materials and Methods

### 2.1. Raw Materials

Snake melon (*C. melo* L. subsp. *melo* Var. *flexuosus* (L.) Naudin) plants of the traditional Spanish accession BGV004853 (held at the UPV genebank) were cultivated in a greenhouse in Meliana (Valencia) (39°31′12.9648″ N, 0°20′30.4044″ W). The plants were directly transplanted into the soil, while water was supplied by drip irrigation. SM fruits (Figure 1) were collected in their commercial state (immature state). Pea protein powder (Nutralys^®^ S85F) (PP) and pea fibre (Pea Fiber I 50 M) (PF) were supplied by Roquette S.L. (Valencia, Spain).

### 2.2. Sample Preparation

SMs (500 g) were washed and homogenised in a Thermomix (TM 21, Vorwerk, Valencia, Spain) for 5 min at 5200 rpm. Once the sample was homogenised, three formulations were prepared by adding 10 g of PP or PF to 90 g of homogenised SMs, and another without biopolymers was used as the control. The formulations were freeze-dried: a puree layer was placed on an aluminium plate and samples were stored at −45 °C (Vertical Freezer, CVF450/45, Ing. Climas, Barcelona, Spain) for 24 h before being dried in a Lioalfa-6 Lyophyliser (Telstar, Terrassa, Spain) at 2600 Pa and −56.6 °C for 48 h. The freeze-dried samples were ground (Minimoka, Taurus, Lleida, Spain) to obtain a free-flowing powder. Therefore, the powdered products obtained from snake melon were SM (snake melon), PPSM (pea protein snake melon), and PFSM (pea fibre snake melon).

### 2.3. Physicochemical and Microstructural Properties

The water content of the powder samples (x_w_) was determined by drying the sample to a constant weight at 70 °C in a vacuum oven [27]. The water activity (a_w_) of the powder samples was determined using the AquaLab PRE (METER Group, Inc. Pullman, WA, USA). The water absorption index (WAI) and water solubility index (WSI) were determined using the method described by Singh and Smith [28]. A 2.5 g sample was dispersed in 25 g of distilled water. After stirring for 30 min using a magnetic stirrer, the dispersions were rinsed into tared 50 mL centrifuge tubes, made up to 32.5 g, and centrifuged at 3000× *g* for 10 min. The supernatant was decanted to determine its dissolved solid content, and the sediment was weighed. WAI and WSI were calculated following García Segovia et al. [20]. To measure hygroscopicity [29], samples (about 1 g in a Petri dish) of each powder were placed at 25 °C in an airtight plastic container containing a Na_2_SO_4_ saturated solution (81% relative humidity) at the bottom. Each sample was weighed after 7 days and hygroscopicity was expressed as g of water gained per 100 g of dry solids.

The powders’ particle size distribution was determined by applying the laser diffraction method and Mie theory, following the ISO13320 regulation [30], using a particle size analyser (Mastersizer 2000, Malvern Instruments Ltd., Malvern, UK) equipped with a wet sample dispersion unit (Hydro 2000 MU, Malvern Instruments Ltd.). Laser diffraction gives the volume of a material of a given size because the light energy reported by the detector system is proportional to the volume of material present. The sample was dispersed in distilled water and pumped through the optical cell under moderate stirring (1800 rpm) at 20 °C. The volume (%) in relation to the particle size (µm) was obtained and the size distribution was characterised by the volume mean diameter (D(4,3)). The standard percentiles d(0.1), or the particle size below which 10% of the sample lay, and d(0.9), or the particle size below which 90% of the sample lay, were also considered for powder characterisation.

To determine the bulk density (ρ_b_), about 2 g of powder was placed inside a 10 mL graduated test tube, and the volume occupied was noted. ρ_b_ was calculated by dividing the mass of powder by the occupied volume and was expressed as g/L. To determine tap density (ρ_T_), a graduated test tube with 2 g of powder was mechanically tapped and volume was recorded until a constant volume was reached. ρ_T_ was calculated by dividing the mass of powder by the occupied volume after tapping and was expressed as g/L. The true density (ρ) of the samples was established with a helium pycnometer (AccPyc 1330, Micromeritics, Norcross, GA, USA). From these determinations, the Hausner ratio (HR), which is correlated with the flowability of a powder [31], was calculated using Equation (1), and the Carr index (CI), which represents the compressibility of a powder [32], was calculated using Equation (2):(1)$$HR=\frac{\rho_{T}}{\rho_{b}}$$
(2)$$CI=100\times \frac{\rho_{T}-\rho_{b}}{\rho_{T}}$$

Moreover, porosity (ε), the percentage of the volume of air related to the total volume, was calculated according to Equation (3) [33]:(3)$$\varepsilon =\frac{\left(\rho -\rho_{b}\right)}{\rho }$$

The morphology and surface microstructures of the powders were examined using a Zeiss Ultra55 Field Emission Scanning Electron Microscope (FESEM; Carl Zeiss AG, Berlin, Germany) with the secondary electron detector (ETSE). The powder was fixed on a carbon adhesive tape and was platinum-coated before analysis. Images were taken at an accelerating voltage of 1 kV and WD 3.5 mm. To examine the microstructure of the samples, the electron mode was used under ×100 magnification. Three representative location areas were imaged for each sample, and at least 12 images at different magnifications (×400, 1000, and 2000) were obtained to ensure the FESEM imaging results were representative.

The colour of the powder samples was measured with a standard D65 illuminate and 10° visual angle (Konica Minolta CM-700d colorimeter, Tokyo, Japan). A reflectance glass (CR-A51, Minolta Camera, Japan) was placed between the sample and colorimeter lens. The measurement window was 6 mm in diameter. The results were expressed using the CIELab system. Chroma (C *, saturation), hue angle (h *), and the total colour difference between the samples with biopolymers (PPSM and PFSM) and SM were measured.

### 2.4. Bioactive Compound Determination

To determine vitamin C (VC), dehydroascorbic acid (DHAA) was reduced to ascorbic acid (AA) using DL-dithiothreitol as a reducing reagent, following Igual et al. [34]. Sample vitamin C content was determined using high-performance liquid chromatography (HPLC) with Jasco equipment (Italy) against the AA standard solution (Panreac, Barcelona, Spain). An Ultrabase-C18 5 mm (4.6 × 250 mm) column (Análisis Vínicos, Tomelloso, Spain), a mobile phase of 0.1% oxalic acid, 20 µL volume injection, and a flow rate of 1 mL/min were used. The detection was undertaken at 243 nm and 25 °C. Results were expressed as mg of AA per 100 g of sample.

Folate content (TF) was analysed using an HPLC-DAD-ESI-MS assay according to the method described by Igual et al. [35]. For sample extraction, 1 g of each sample was homogenised with 5 mL of phosphate buffer (pH = 7) and sonicated using an ultrasonic bath (Elmasonic E15H, Elma Schmidbauer GmbH, Singen, Germany). The obtained solution was further centrifuged at 4000× *g* for 10 min at 24 °C, filtered using a nylon filter (0.45 µm, Millipore Merck KGaA, Darmstadt, Germany), and injected into the XDB C18 Eclipse (4.5 × 150 mm, particle size: 5 µm) HLPC column. The column parameters were acetonitrile:acetic acid 1% in a ratio of 20:80 (*v*/*v*), flow rate of 0.5 mL/min, and temperature of 25 °C. For the MS fragmentation, the capillary voltage was set at 300 V, and a nitrogen flow of 7 L/min and a scanning range of 120–600 *m*/*z* in the ESI (+) mode were applied. Agilent ChemStation software (Rev B.04.02 SP1) was used for data acquisition, and chromatograms were recorded at λ = 280 nm. A standard folic curve (y = 126.25x − 16.283, r^2^ = 0.9945) with a range of 30 µg/mL to 1 µg/mL was used.

Total carotenoids (TCs) were extracted with a solvent hexane/acetone/ethanol mixture following the method described by Olives-Barba et al. [36]. The spectrophotometric reference method provided by the AOAC (2000) was used for quantification [37]. The absorbance was measured at 446 nm in a UV-3100PC spectrophotometer (VWR, Leuven, Belgium). The TC content was expressed as mg of β-carotene (Sigma-Aldrich, Steinheim, Germany) per 100 g of sample. From the TC extract, chlorophyll a was determined. Sample absorbance was measured at 663 nm in a UV-3100PC spectrophotometer (VWR, Leuven, Belgium). The chlorophyll a content was expressed as mg/100 g of sample and calculated following Lichtenthaler and Buschmann [38] and Zvezdanovic and Markovic [39].

Determination of total phenols (TPs) was based on the Folin–Ciocalteu method. The extraction procedure involved mixing the sample with methanol. The mixture was centrifuged (12,857× *g*, 10 min, 4 °C) to obtain the supernatant [34]. Absorbance was measured at 765 mm in a UV-3100PC spectrophotometer (VWR). The total phenolic content was expressed as mg of gallic acid (Sigma-Aldrich, Steinheim, Germany) equivalents (GAE) per 100 g.

The antioxidant capacity (AC) was assessed using the free radical scavenging activity of the samples as evaluated with the stable radical 2,2-diphenyl-1-picryl-hydrazyl-hydrate (DPPH) (Sigma-Aldrich) following the methodology described by Igual et al. [34] in triplicate. Samples were mixed with methanol and the homogenate was centrifuged (12,857× *g*, 10 min, 4 °C) to obtain the supernatant. The supernatant (0.1 mL) was added to 3.9 mL of DPPH (0.030 g/L) in methanol. A UV-3100PC spectrophotometer (VWR) measured the absorbance at 515 nm. The results were expressed as milligram Trolox equivalents (TE) per 100 g.

All bioactive compounds of the samples were analysed in triplicate.

### 2.5. Encapsulation Efficiencies (EEs)

To evaluate the EE, the total phenols (TPs) or antioxidant capacity (AC), depending on the case (represented as TB), and the analysed surface bioactive compounds (SB) for the samples were determined after freeze-drying [20]. To determine TB, samples were treated according to TPs or AC. To determine SB, the samples were not ground to destroy the microcapsules but were only extracted with the solvents in a vortex for 30 s and filtered through a 0.45 µm size filter following the procedure described by Idham et al. [40]. The EE% was calculated using Equation (4):(4)$$\%EE=\frac{\left(TB-SB\right)}{TB}\times 100$$

### 2.6. Statistical Analysis

Analysis of variance (ANOVA) was applied with a confidence level of 95% (*p* < 0.05) to evaluate the differences among samples. Furthermore, a correlation analysis of the studied bioactive compounds and antioxidant capacity of the powders was conducted with a 95% significance level. Statgraphics Centurion XVII software, version 17.2.04 (Statgraphics Technologies, Inc., The Plains, VA, USA), was used.

## 3. Results

### 3.1. Physicochemical and Microstructural Properties of SM Powders

The parameters x_w_, a_w_, WAI, WSI, and Hy provide information about the stability of powdered products and are shown in Table 1. Samples with biopolymers in their formulation (PPSM and PFSM) showed values for x_w_ and a_w_ that were significantly lower than those for SM (without biopolymers). Therefore, the incorporation of PP and PF significantly reduced (*p* < 0.05) the water content in the powder and the available water in the powder product, thus resulting in samples that were more resistant to degradation. Moisture content is a property of powder related to the drying efficiency, powder flowability, stickiness, and storage stability due to its effect on glass transition and crystallisation behaviour [41]; thus, SM would be more susceptible to undesirable physical changes. WAI and WSI are two indices that show the behaviour of the powdered product in contact with water. The WAI indicates the amount of immobilised water in the samples [42], whereas the WSI relates to the soluble solids in the product as a function of the solubilisation of starches, sugars, proteins, fibres, and maltodextrin [43]. The incorporation of PP or PF in SM powders significantly reduced the WAI (*p* < 0.05), thus reducing the immobilised water in the sample. However, the addition of PP significantly decreased the WSI (*p* < 0.05), providing stability and reducing degradation due to solubilisation of the compounds; however, with PF, the opposite trend was observed. PF partly contains soluble fibre, which contributed to its solubilisation and made the powder more susceptible to degradation. There are studies that have observed the same trends in powders obtained by spraying beetroot with pea protein (a reduction in the WSI [20]) and in powders obtained by spraying orange juice with resistant maltodextrin, which is a soluble fibre that also caused an increase in the WSI of the powder [33]. SM contains sugars (mainly glucose and fructose) and organic acids (mainly malic but also citric and glutamic) in its composition [5], and when it becomes a powder, low-molecular-weight sugars and organic acids—with low Tg and moisture content—cause high hygroscopicity [44]. In this study, the Hy of SM was significantly higher than that of the powders with biopolymers (*p* < 0.05). The addition of high-molecular-weight solutes favours the attenuation of this phenomenon [45,46]. PPSM and PFSM showed significant differences (*p* < 0.05), but the values were in the same range.

Particle size distribution is directly related to the physical properties of a powdered product. Properties such as bulk density, compressibility, and flowability depend strongly on the particle size and its distribution [47]. Figure 2 shows the volume–particle size distributions for the three samples studied. SM presented a distribution that differed markedly from the other samples, with a higher volume percentage of larger particle sizes. Here, PPSM presented a higher volume of larger particles, whereas PFSM showed a higher volume of particles in the range 3–50 µm. From these volumetric particle size distributions, the D(4,3), d(0.1), d(0.5), and d(0.9) for the three powder products were studied and are shown in Table 2. There were significant differences among the samples (*p* < 0.05). The PFSM showed a smaller particle size (D) than the other samples, followed by the PPSM. The same trend was observed for d(0.1), d(0.5), and d(0.9). However, the bulk density and porosity showed opposite trends (Table 2); SM had the lowest ρ_b_ and the highest ε, whereas PFSM had the highest ρ_b_ and lowest ε. Porosity plays a key role in the agglomeration strength of dried foods [48]. Furthermore, higher porosity (and lower bulk density) corresponds to a greater air volume distributed among particles, and the water inlet could be more accessible [48,49]. A greater volume of air among particles is conditioned by a larger particle size because, when larger particles settle, they leave air spaces of a greater volume than when the particles are smaller. Smaller particle size allows for better organisation of particles and leaves smaller gaps between particles, therefore resulting in less porosity, as with PFSM. Table 2 also shows the HR and CI. HR is an index correlated with the flowability of a powder. The range for HR in defining flowability is as follows [50]: free-flowing powder (1.0 < HR < 1.1), medium-flowing powder (1.1 < HR < 1.25), difficult-flowing powder (1.25 < HR < 1.4), and very difficult-flowing powder (HR > 1.4). According to this ranking, PPSM and PFSM were medium-flowing powders but near to free-flowing powders. However, the SM was a difficult-flowing powder. The CI represents the compressibility of a powder. According to Carr, excellent flowability can be expected if the CI is within 5 to 15%, as with PPSM and PFSM. The SM showed about 21%, which is an intermediate value. Values for the CI above 25% indicate poor flowability [30]. PPSM and PFSM did not show significant differences in the HR and CI (*p* > 0.05).

Table 3 shows the Pearson correlation coefficients for the different physicochemical parameters of the powder products studied. All parameters showed significant correlations with other parameters (*p* < 0.05), except the WSI. The WAI presented highly significant Pearson coefficients with all the parameters studied except the WSI. The D(3.4), d(0.1), d(0.5), and d(0.9) showed the highest correlation coefficients with the WAI, as well as ρ_b_ and ε. The particle size of the powdered products directly affected the water absorption capacity in a linear and positive way. A close positive relationship between ε and the WAI and a negative relationship between ρ_b_ and the WAI were also observed. Table 3 highlights the effect of particle size on other powder properties. A larger powder particle size led to higher porosity and hygroscopicity and lower ρ_b_ according to the Pearson coefficients. However, the flowability of the powder decreased (increase in the HR and CI) when the water content, water activity, and hygroscopicity of the powder increased.

Figure 3 shows FESEM micrographs of the SM, PPSM, and PFSM samples. SM showed particles with irregular shapes, pores, and wrinkles (Figure 3A,D,G). This was similar to the findings of other authors, who explained the particle morphology by the low temperature used in the freeze-drying process, which resulted in a lack of forces to break up the frozen liquid into droplets [51]. Using PP and PF during freeze-drying makes it possible to obtain more regular and defined particles. PPSM (Figure 3B,E,H) showed more wrinkled particles than PFSM (Figure 3C,F,I), which presented oval or spherical shapes with smooth-surfaced particles. PFSM powdered particles were smaller, with a higher particle density observed in the analysed field. This agrees with the results from Table 2, which indicate smaller particle sizes and lower ε in PFSM. As indicated by Figure 3, PPSM and PFSM showed greater encapsulation because they formed defined particles, which was more remarkable in the PFSM.

The appearance of the studied SM powders is shown in Figure 4. Visual differences can be seen in both the colour and flow of the powders. The powders with PP or PF were less green but more fluid, as indicated in Table 2. Furthermore, SM appeared to form small aggregates and was more hygroscopic. Despite the differences, the three powders had good appearances. PPSM and PFSM stood out for having more powder-like appearances.

Table 4 shows the colour coordinates and the total colour differences (ΔE *) for the SM powders. As indicated by the colour coordinates, SM presented greenish tones (negative a * value), as shown in Figure 4, corresponding to the tone of the raw material (Figure 1). PP and PF addition significantly reduced the a *, b *, and C * (*p* < 0.05), whereas h * and L * increased. Therefore, PPSM and PFSM were whiter powders and less green than the powder without biopolymers. Other studies have explained that adding a carrier agent (maltodextrin, gum Arabic, or maltodextrin-resistant starch) can cause whiter powders because these carrier agents are white [33,52,53]. The PFSM was significantly whiter and greener than the PPSM (*p* < 0.05). The PFSM also showed significantly lower values for b * and C * (*p* < 0.05) but higher values for h * compared to the PPSM. The total colour differences between samples with biopolymers and the SM were higher than 3 units (Table 4). Therefore, they were perceptible by the human eye, which only distinguishes colour difference if ΔE * is higher than 3 [54]. There were significant colour differences between the PPSM and PFSM. The greatest colour differences were observed in powders with PF.

### 3.2. Bioactive Compounds and Efficiency Encapsulation

As the samples with biopolymers had different amounts of SM, the content of the bioactive compound in the powders was considered in relation to the SM’s own solutes to evaluate the effect of biopolymer addition in bioactive compounds. Table 5 shows the mean values and standard deviations for the content of the bioactive compounds and the AC expressed in mg/100 g_solutesSM_. The AA content of the samples showed significant differences (*p* < 0.05), with higher AA content in the PPSM. For VC, the samples with biopolymers showed significantly higher values, especially the PFSM. The difference between the VC and AA content indicated the DHAA content, which also has a vitamin function but is a degraded form of AA [55]. The TF, TC, and Cha were significantly higher in PFSM compared to the other powders. However, TPs content differed among samples, but the three powders were within the same statistically homogeneous group (*p* > 0.05). Finally, AC was significantly higher in the samples with biopolymers (PPSM and PFSM (*p* < 0.05)). Biopolymers have been added during the production of food powders to act as encapsulating or wall materials, helping to keep the desired functional properties in the finished product, such as stability against oxidation, ease of handling, improved solubility, controlled release, and extended shelf life [56]. According to the results referring to the SM’s own solutes, a protective effect from PP and PF was observed. Furthermore, PF stood out because it showed significant differences (*p* < 0.05) compared to PP in protection against the degradation of some of the bioactive compounds studied and the loss of AC. Another study also showed improvements in the protection of bioactive compounds, especially VC and flavonoids, with the addition of bamboo fibre to freeze-dried grapefruit juice [57]. A further work with rose hip presented PP as the best protective encapsulation agent for the production of powdered products by freeze-drying [21].

To explain the relationships in the bioactive compounds with AC, correlation statistical analyses were performed. Vitamin C and TPs played key roles for the AC of SM powders, showing Pearson coefficients of 0.9736 (*p* < 0.05) and 0.8624 (*p* < 0.05), respectively. This behaviour has also been observed by other authors in vegetable products [33,50].

Figure 5 shows the TPs and AC EE% for the SM freeze-dried samples. The EE represents the potential of the wall material to encapsulate or hold the core material inside the microcapsule [38]. The EE is also related to the shelf life of the phenolic compound content and the AC in the powder. The low EE% for the TPs and AC of the powder without biopolymers stood out, but this behaviour was expected considering that it did not contain carrier agents encapsulating the bioactive compounds in its formulation. The PPSM EE% values for TPs and AC were like those obtained with other vegetable powders encapsulated with pea protein [24]. PFSM presented the highest EE% for TPs and AC, showing significant differences compared to PPSM (*p* < 0.05). Relating these results to Figure 3, the smooth-surfaced oval or spherical particles were probably a more consistent barrier to bioactive compound degradation than with PPSM, which presented wrinkles on the surface and more irregular shapes.

## 4. Conclusions

Powdered SM products are a suitable alternative to the whole fruit, facilitating transportation and avoiding seasonality issues, because they are stable products due to low water activity. Furthermore, their consumption would increase if they were rehydrated as juices or infusions; added to desserts, dairy products, salads, ice creams, and snacks; or used to enrich food with bioactive compounds. Increasing SM demand can contribute to promoting its cultivation, which would be especially interesting for organic farming practices in agricultural regions affected by soil problems.

SM powders contain vitamin C, folates, carotenoids, chlorophyll a, and phenols, which contribute to their AC. Incorporating PP or PF in the formulation before freeze-drying generated stable encapsulates that protected the bioactive compounds. PPSM and PFSM were less hygroscopic and more free-flowing and had lower water content and water activity compared to SM. PFSM showed higher efficiency in encapsulation and smaller particles with a smooth surface and oval shape. However, it showed the greatest differences in colour from SM due to the white tone of the fibre.

## Figures and Tables

**Figure 1 foods-12-02679-f001:**
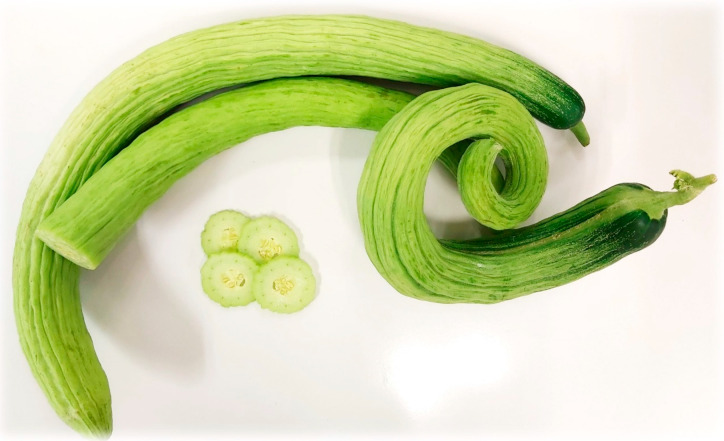
Snake melon fruit and transverse section of the fruit.

**Figure 2 foods-12-02679-f002:**
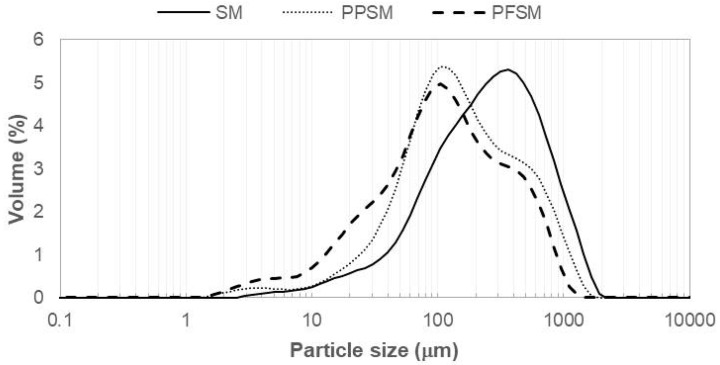
Volume–particle size distributions (representative curves) for obtained powders. SM, snake melon; PPSM, snake melon with pea protein; PFSM, snake melon with pea fibre.

**Figure 3 foods-12-02679-f003:**
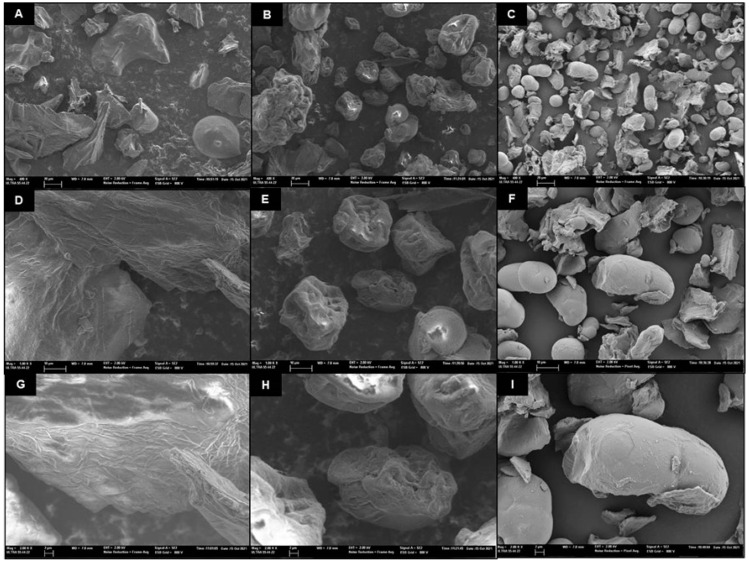
FESEM micrographs with ×400 (**A**–**C**), ×1000 (**D**–**F**), and ×2000 magnifications of the studied powders: snake melon (**A**,**D**,**G**), snake melon with pea protein (**B**,**E**,**H**), and snake melon with pea fibre (**C**,**F**,**I**).

**Figure 4 foods-12-02679-f004:**
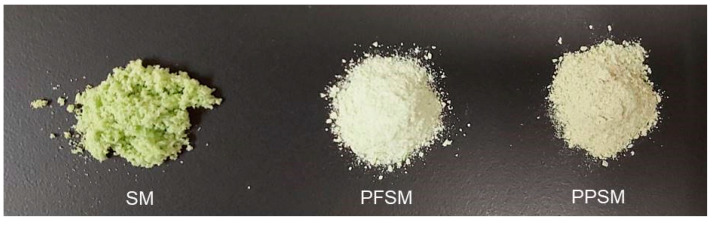
Appearance of the studied powders. SM, snake melon; PPSM, snake melon with pea protein; PFSM, snake melon with pea fibre.

**Figure 5 foods-12-02679-f005:**
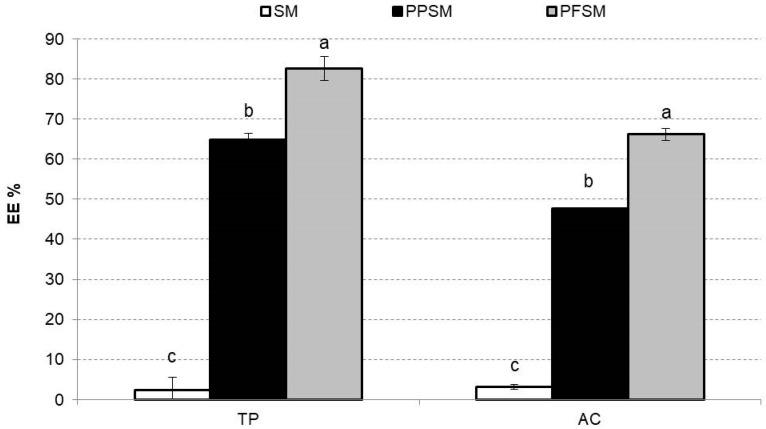
Mean values and standard deviations for encapsulation efficiency percentages in snake melon powders for total phenols (TPs) and antioxidant capacity (AC). Letters (a–c) indicate homogeneous groups established by the ANOVA (*p* < 0.05) for TPs or AC. SM, snake melon; PPSM, snake melon with pea protein; PFSM, snake melon with pea fibre.

**Table 1 foods-12-02679-t001:** Mean values (and standard deviations) for water content (x_w_), water activity (a_w_), the water absorption index (WAI), the water solubility index (WSI), and hygroscopicity (H_y_) for snake melon powders.

	SM	PPSM	PFSM
x_w_ (g_w_/g_sample_)	0.048 (0.014) ^a^	0.003 (0.002) ^b^	0.006 (0.004) ^b^
a_w_	0.225 (0.003) ^a^	0.031 (0.03) ^b^	0.039 (0.003) ^b^
WAI (g_sed_/100 g_dry solid_)	9.075 (0.012) ^a^	6.4 (0.3) ^b^	5.36 (0.18) ^c^
WSI (%)	24.4 (0.2) ^b^	16.2 (0.9) ^c^	29.0 (0.6) ^a^
H_y_ (g_w_/100 g_dry solid_)	106.2 (1.5) ^a^	58.40 (0.06) ^b^	53.9 (0.2) ^c^

The same letters in superscript within rows indicate homogeneous groups established by ANOVA (*p* < 0.05). SM: snake melon; PPSM: pea protein snake melon; PFSM: pea fibre snake melon.

**Table 2 foods-12-02679-t002:** Mean values (and standard deviations) for volume mean diameter D(4,3); standard percentiles d(0.1), d(0.5), and d(0.9); bulk density (ρ_b_); the Hausner ratio (HR); the Carr index (CI); and porosity (ε) for snake melon powders.

	SM	PPSM	PFSM
D(4,3) (μm)	340 (22) ^a^	230 (21) ^b^	163 (14) ^c^
d(0.1) (μm)	50.1 (1.4) ^a^	32.5 (1.2) ^b^	16 (2) ^c^
d(0.5) (μm)	242 (11) ^a^	132 (7) ^b^	95 (7) ^c^
d(0.9) (μm)	781 (63) ^a^	587 (62) ^b^	430 (39) ^c^
ρ_b_ (g/cm^3^)	0.1060 (0.0007) ^c^	0.253 (0.002) ^b^	0.323 (0.009) ^a^
HR	1.260 (0.014) ^a^	1.151 (0.012) ^b^	1.1765 (0.0012) ^b^
CI	20.6 (0.9) ^a^	13.1 (0.9) ^b^	15.0 (0.2) ^b^
ε	0.9333 (0.0013) ^a^	0.835 (0.002) ^b^	0.766 (0.002) ^c^

The same letters in superscript within rows indicate homogeneous groups established by ANOVA (*p* < 0.05). SM: snake melon; PPSM: pea protein snake melon; PFSM: pea fibre snake melon.

**Table 3 foods-12-02679-t003:** Pearson correlation coefficients for physicochemical parameters of studied powders.

	a_w_	WAI	WSI	H_y_	D(4,3)	d(0.1)	d(0.5)	d(0.9)	ρ_b_	HR	CI	ε
x_w_	0.9552 *	0.9061 *	0.2202	0.9433 *	0.8782 *	0.7991 *	0.9168 *	0.8376 *	−0.8820 *	0.9029 *	0.9024 *	0.8500 *
a_w_		0.9471 *	0.1956	0.9932 *	0.9340 *	0.8567 *	0.9599 *	0.8699 *	−0.9359 *	0.9704 *	0.9666 *	0.8957 *
WAI			−0.1237	0.9761 *	0.9905 *	0.9655 *	0.9925 *	0.9711 *	−0.9922 *	0.8544 *	0.8450 *	0.9877 *
WSI				0.0830	−0.1589	−0.3276	−0.0770	−0.2918	0.1559	0.3756	0.3932	−0.2583
H_y_					0.9677 *	0.9090 *	0.9836 *	0.9155 *	−0.9691 *	0.9454 *	0.9397 *	0.9403 *
D(4,3)						0.9844 *	0.9954 *	0.9831 *	−0.9999 *	0.8430 *	0.8337 *	0.9926 *
d(0.1)							0.9649 *	0.9912 *	−0.9835 *	0.7398	0.7281	0.9936 *
d(0.5)								0.9725 *	−0.9960	0.8758 *	0.8679 *	0.9814 *
d(0.9)									−0.9830 *	0.7399	0.7289	0.9906 *
ρ_b_										−0.8443 *	−0.8349 *	−0.9930 *
HR											0.9997 *	0.7850
CI												0.7741

Water content (x_w_); water activity (a_w_); water absorption index (WAI); water solubility index (WSI); hygroscopicity (H_y_); volume mean diameter D(4,3); standard percentiles d(0.1), d(0.5), and d(0.9); bulk density (ρ_b_); Hausner ratio (HR); Carr index (CI); and porosity (ε). * Correlation was significant at 0.05.

**Table 4 foods-12-02679-t004:** Mean values (and standard deviations) for colour coordinates (L *, a *, b *, C *, and h *) and total colour differences (ΔE) for snake melon powders.

	SM	PPSM	PFSM
L *	70.6 (0.5) ^c^	75.3 (0.2) ^b^	79.4 (0.7) ^a^
a *	−9.42 (0.03) ^c^	−2.92 (0.07) ^a^	−6.69 (0.13) ^b^
b *	28.65 (0.15) ^a^	25.1 (0.2) ^b^	20.8 (0.4) ^c^
C *	30.16 (0.13) ^a^	25.26 (0.18) ^b^	21.9 (0.4) ^c^
h *	288.20 (0.13) ^a^	276.6 (0.2) ^c^	287.82 (0.18) ^b^
ΔE		8.795 (0.105) ^b^	12.1 (0.6) ^a^

The same letters in superscript within rows indicate homogeneous groups established by ANOVA (*p* < 0.05). SM: snake melon; PPSM: pea protein snake melon; PFSM: pea fibre snake melon.

**Table 5 foods-12-02679-t005:** Mean values (and standard deviations) for ascorbic acid (AA), vitamin C (VC), total folates (TF), total carotenoids (TCs), chlorophyll a (Ch_a_), total phenols (TPs) content, and antioxidant capacity (AC) for snake melon powders. Units: mg/100 g_solutesSM_.

	SM	PPSM	PFSM
AA	25.91 (0.09) ^b^	28.9 (0.3) ^a^	25.52 (0.03) ^c^
VC	54.86 (0.09) ^c^	65.3 (0.9) ^b^	70.63 (0.12) ^a^
TF	1.038 (0.003) ^b^	1.12 (0.12) ^b^	1.44 (0.06) ^a^
TC	32.85 (0.13) ^b^	32.05 (0.09) ^c^	40.7 (0.3) ^a^
Ch_a_	84.7 (0.9) ^b^	86.5 (0.7) ^b^	99. (2) ^a^
TP	258 (5) ^b^	291 (21) ^ab^	307 (4) ^a^
AC	186 (9) ^b^	229 (2) ^a^	240 (4) ^a^

The same letters in superscript within rows indicate homogeneous groups established by ANOVA (*p* < 0.05). SM: snake melon; PPSM: pea protein snake melon; PFSM: pea fibre snake melon.

## Data Availability

The data presented in this study are available on request from the corresponding author. The data are not publicly available due to next works are ongoing.

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
