# Peer review of "Physicochemical, Structural, and Functional Properties of Snake Melon (Cucumis melo subsp. melo Var. flexuosus) Microencapsulated with Pea Protein and Pea Fibre by Freeze-Drying"

_foods, 2023, doi:10.3390/foods12142679_

Round 1

Reviewer 1 Report

Introduction 

Why did you use pea protein and pea fibre to encapsulate the powder?

2. What has been previously done ? Use it to justify this work.

Methodology 

How did you select 10 g  on line 87?

What might be the reason that fibre and protein have a lower WAI compared to melon powder?

 From literature what is pea fibre and pea protein composed of which may make it less hydrophilic? Does pea fibre form a gel in water? I see it's high solubility!

Table 5. SM looks green and also has a highest negative a value for colour but has the least Chlorophyll a? What might be the reason? The units are confusing, why not mg/100 g powder? 

Adding pea prqotein and pea fibre increased the vitamin c and folate content? What was the purpose of analysing folates? Are they high in SM? So the pea protein and fibre have even higher?

English language is ok.

Author Response

The reply is attached.

Reviewer 2 Report

The aim of the research described in the manuscript is to compare the physicochemical and structural parameters of lyophizates obtained from Snake melon and protein and fiber preparations. The topic is interesting and practical. It is a pity that the storage experiment was not carried out in the field of research. Many analytical methods are used in the article and they are correct and appropriate. In my opinion, only the research material is quite small to draw deep conclusions. Only three samples were tested. I believe that such a statement should be included in the final conclusions that the research requires confirmation, e.g. on different doses of the addition of preparations.

The introduction does not present the innovativeness of the research topic in scientific terms. It was perfectly explained why such a research topic was started in terms of its practical application. However, there is no information whether studies on the use of freeze-drying with various additives have been carried out before and what conclusions were drawn from the use of these processes. If no such research has been conducted, it should also be written in the text to emphasize the innovative nature of the research.

The addition of Pea protein powder 79 was used in the work (Nutralys® S85F) (PP) and pea fiber (Pea Fiber I 50 M). The manufacturer's information should be included on the packaging of these raw materials.

The content of vitamin C was determined in the work. Was the degradation of this vitamin protected in any way during sample preparation. Homogenization was applied, then preparations were added and the sample was frozen. Was the preparation time always the same? Was it observed that due to the determination of compounds not resistant to oxidation processes?

How was the formulation of the preparations developed? The addition of proteins and fiber was as suggested by its manufacturer, whether preliminary research was done. If so, what parameters determined the choice of dose?

The lyophilization process is considered one of the technological processes that does not change the nutritional value of the products. Only it is not recommended to protect products containing carotenoids. In the tested lyophilisates, it was shown that the addition of protein and fiber resulted in lower losses of active compounds than in lyophilisates without these preparations. Do you know what losses the lyophilization process caused? Even when compared to the literature, why were there such large differences? And were losses not taken into account during the lyophilization process itself, but before, i.e. preparation of preparations and freezing? Were the preparations sufficiently purified that they did not contain ingredients from other plants, e.g. polyphenols? The more so that the addition of proteins and fiber diluted the polyphenols contained in the test material. The discussion on this topic should be more extensive. The results should be compared to other publications.

Author Response

The reply is attached.

Reviewer 3 Report

Dear Editor and Authors,

The manuscript ‘Physicochemical, structural, and functional properties of Snake  melon (Cucumis melo subsp. melo Var. flexuosus) microen-  capsulated with pea protein and pea fibre by freeze-drying’ by Marta Igual 1*, Alejandro Flores-León 2 , Belén Picó 2 , Javier Martínez-Monzó 1 and Purificación García-Segovia describes research on snake melon freeze-dried powders with or without biopolymers (pea  protein (PPSM) or pea fibre (PFSM). The Authors studied the physicochemical, structural, and functional properties of  the powders. The study is interesting, well planned, methods were selected properly, references cited properly, conclusions made based on the results.

I have found some minors that should be corrected:

Line 29 Latin names in italics

Line 84 what was the mass (weight) of SM

Line 159 give more details on vitamin C determination conditions: column, flow etc..

Table 1 and further

Tables should be self-explanatory. Give full names of the preparations in the title, in preparation names-no abbreviations or give a legend below the table.

That is why I recommend minor revision,

Yours sincerely,

Author Response

The reply is attached.

Round 2

Reviewer 2 Report

My comments have been incorporated into the revised manuscript.